# Environmental Microbial Contamination during Cystic Fibrosis Group-Based Psychotherapy

**DOI:** 10.3390/ijerph18031142

**Published:** 2021-01-28

**Authors:** Martina Rossitto, Paola Tabarini, Vanessa Tuccio Guarna Assanti, Enza Montemitro, Arianna Pompilio, Ersilia Vita Fiscarelli

**Affiliations:** 1Cystic Fibrosis Diagnostic Unit, Laboratory and Specialistic Pediatrics Departments, Bambino Gesù Children’s Hospital, IRCCS, 00165 Rome, Italy; martina.rossitto@opbg.net (M.R.); vanessa.tuccio@opbg.net (V.T.G.A.); 2Neuroscience and Neurorehabilitation Department, Bambino Gesù Children’s Hospital, IRCCS, 00165 Rome, Italy; paola.tabarini@opbg.net; 3Cystic Fibrosis Center, Specialistic Pediatrics Department, Bambino Gesù Children’s Hospital, IRCCS, 00165 Rome, Italy; enza.montemitro@opbg.net; 4Center for Advanced Studies and Technology (CAST), “G. d’Annunzio” University of Chieti-Pescara, 66100 Chieti, Italy; arianna.pompilio@unich.it; 5Department of Medical, Oral and Biotechnological Sciences, “G. d’Annunzio” University of Chieti-Pescara, 66100 Chieti, Italy

**Keywords:** cystic fibrosis, group-based psychological/psychoanalytic interventions, microbiological environmental monitoring

## Abstract

Living with cystic fibrosis (CF) exposes patients to the risk of developing anxiety and depression, with therapeutic compliance reduction, hospitalization increase, and quality of life and health outcomes deterioration. As pulmonary infections represent the major cause of morbidity and mortality in patients with CF, environmental contamination due to droplet dispersion and the potential transmission from environment to such patients should be prevented. Therefore, in-person contact, including group-based psychotherapy, are strongly discouraged. Nevertheless, group sharing of disease-related experiences represents a way to recover the inner resources essential for dealing with a chronic pathology. Keeping in mind the guidelines for infection control, the aim of this study is to evaluate the risk of the dissemination of microorganisms in a restricted environment where patients with CF attend group psychotherapy sessions. Five patients, selected according to their microbiological status, attended 32 group-based psychological/psychoanalytic meetings. Before each session, they were asked to observe the infection control recommendations. Microbiological environmental monitoring (MEM) has been performed to evaluate both air and surface contamination. As reported, a strict observation of standard precautions allows one to avoid environmental contamination by pathogens of the CF respiratory tract. Although infection control guidelines discourage group-based psychological/psychoanalytic interventions, our observations report the feasibility and safety of group psychotherapy when strict precautions are taken.

## 1. Introduction

Cystic fibrosis (CF) is a chronic life-shortening genetic disease affecting several organs, primarily lung and pancreas, as well as entire systems, namely the reproductive and digestive. As a result of the remarkable improvements of the past decades (from new emphasis on early diagnosis to important new treatments, with an understanding of the CF clinical spectrum), CF is no longer solely a childhood disease [1]. In many countries, adults with CF have outnumbered pediatric patients, with a longer average life expectancy for patients born between 2012 and 2016, equal to 43 years in the United States and 53 in Canada [2]. However, patients with CF have to undergo numerous and time-consuming daily treatments to reach a good life expectancy; a burden that, associated with a potentially declining health, may worsen their quality of life [3]. Patients with a chronic disease, as they get older, are exposed to the risk of developing symptoms, such as anxiety and/or depression, jeopardizing their adherence to therapeutic treatments and deteriorating their physical and social functionality (e.g., worse pulmonary function and interpersonal relationships). In this regard, several studies have reported higher rates of depression in patients with CF compared to the healthy population [4,5,6,7]. 

Patients with chronic diseases (e.g., diabetes, cancer) are often involved in summer and educational camps or support groups to improve their quality of life, facilitate psychological adaptation and stimulate adherence to treatment through sharing the illness experiences between peers [3]. In addition, group-based psychological/psychoanalytic interventions, in which, supported by a psychotherapist, each participant becomes the other’s therapist in a sharing interaction not achievable in individual therapy, become crucial in chronic conditions. During these group-based psychotherapy sessions, participants can share disease-related physical and psychological problems, helping them to empathize, get feedback, and ultimately, through the relationship with each other, transform the problematic patterns [8]. 

Unfortunately, the organization of group-based psychological interventions for CF patients is discouraged by the current infection control guidelines implemented in CF care settings [3]. CF airways could be chronically colonized by some bacterial pathogens, in particular *Pseudomonas aeruginosa* (*P. aeruginosa*), *Staphylococcus aureus* (*S. aureus*), *Burkholderia cepacia* complex, and *Stenotrophomonas maltophilia* (*S. maltophilia*), among others. All the above microorganisms can be transmitted from one patient to another, through a combination of direct and indirect contact with infectious secretions, e.g., by objects or airflows that are contaminated (i.e., via droplets or droplet nuclei). Generated by coughing and sneezing, droplets remain suspended in the air for a few minutes, being able to travel as far as 2 m, directly reaching the recipient respiratory tract, as well as horizontal environmental surfaces and patients’ hands [9]. Furthermore, the smaller nuclei resulting from droplet evaporation are capable of airborne transmission, traveling through the air for longer periods and distances, thus allowing pathogens to spread even among individuals not sharing a common air space [10]. Several studies have demonstrated that coughing can generate droplet nuclei containing aerosolized common CF pathogens (i.e., *Achromobacter* spp., *Burkholderia* spp., *S. aureus*, *S. maltophilia*, *P. aeruginosa*, and *Mycobacterium abscessus*), which can travel up to 4 m, and surviving within the droplet nuclei for up to 45 min [10,11,12,13]. Person-to-person transmission of pathogens such as *Burkholderia* spp. and *P. aeruginosa* has been described both in non-healthcare setting (e.g., summer campus) [14] and in CF care centers [9]. Therefore, the infection control guidelines strongly discourage in-person contacts among subjects with CF, and recommend wearing surgical masks and keeping a distance of at least 1 m [15] in all settings to reduce the risk of CF pathogen transmission through droplets [9].

Although there is no evidence on CF typical pathogen transmission related to educational program attendance [16], psychosocial care, now well-integrated into the CF clinical path [1], does not take into account group-based psychological interventions, leading to potential social isolation [4]. Therefore, the aim of this work is to investigate whether a restricted number of patients with CF may effectively contaminate the environment that hosts group-based psychological/psychoanalytic sessions with CF-common pathogens, with a particular focus on *P. aeruginosa*, despite adoption of all the precautions recommended by the infection control guidelines.

## 2. Materials and Methods 

From 2013 to 2015, group-based psychological/psychoanalytic interventions were organized for subjects with CF in follow-up at the Bambino Gesù Children’s Hospital CF Center (Rome, Italy). The patients involved, two males and three females, had median values Forced Expiratory Volume in 1 second (FEV1) of 46.5% ± 19.2 (range: 41–99%) and were aged between 17 and 27 years (average age 21 ± 4.55). Moreover, as for patients’ enrollment, their microbiological status was considered (Table 1).

According to our CF Center’s segregation policy, subjects were excluded if colonized by *B. cepacia* complex, methicillin-resistant *S. aureus* or nontuberculous mycobacteria (NTM), whereas patients were considered eligible when chronically colonized by multidrug-resistant *P. aeruginosa*.

In line with the infection control guidelines in force in our CF Center [15], the patients could attend meetings only if wearing a surgical mask and disposable gown, performing hydroalcoholic hand disinfection before and after every session, and maintaining at least a 1 m distance from each other. [9] All subjects involved provided written informed consent. The study was conducted following the ethical principles of the Declaration of Helsinki and the national laws.

The psychological group-based sessions were held in a 28 m^2^ room. Air conditioning was turned off each time while the two windows were left partially open in order to guarantee minimal ventilation without altering the environmental measurements. A table, 0.50 m high, 0.80 m wide and 0.80 m deep, was placed in the middle of the room and cleaned prior to its use with hydroalcoholic surface disinfectant. All participants and a psychologist sited keeping at least a 1 m distance from each other and from the table.

Microbiological environmental monitoring (MEM) was performed to evaluate air and surface contamination. Air contamination was evaluated through passive sampling [18] by quantifying the microbial fall-out onto five petri dishes containing Trypticase Soy Agar (TSA; Thermo Fisher Diagnostic, Rodano, Italy) according to the 1/1/1 scheme with some modifications. The plates were positioned at the center and the edges of the table (i.e., at 0.5 m from the floor instead of 1 m as indicated in the scheme), left open facing upwards to the air for 1 h, and at least 1 m away from any obstacle (i.e., participants) [19]. The last condition was modified after analyzing the results of the first five meetings, allowing one patient at a time not to wear the surgical mask and reducing his/her distance from the table from 1 to 0.5 m. In this way, the closest TSA plate was 0.5 m from the patient, whereas the furthest was more than 1 m. The patient without a mask was encouraged to observe cough etiquette, covering their mouth with the hand or arm when coughing. 

The microflora responsible for surface contamination was monitored by using RODAC (replicate organism detection and counting) contact petri plates (Liofilchem srl, Roseto degli Abruzzi, Italy) for 10 s in three determined spots at two specific times: (i) right after its cleaning, before patients entered the room; and (ii) at the end of the psychological sessions. 

TSA and RODAC plates were incubated for 48 h at 37 ± 2 °C under aerobic conditions. Colonies with different morphotypes were isolated on Columbia agar +5% sheep blood (bioMérieux, Marcy l’Etoile, France), and subsequently identified by MALDI-TOF mass spectrometry (Bruker, Billerica, MA, USA). On the basis of their ecological behavior [20], the bacteria were classified into two groups: (i) potentially human-derived bacteria, and (ii) truly environmental bacteria. Total microbial load was expressed as CFU/m^2^/h and CFU/cm^2^ for air and surface contamination, respectively. 

## 3. Results

### MEM Results

Based on their microbiological status, five CF patients (Pt1–Pt5) were selected as the group participants. All subjects involved attended 32 group-based psychological/psychoanalytic sessions, with the exception of Pt2 who, unfortunately, died at the end of 2014, and therefore participated in only 21 sessions. 

The bacterial species recovered through MEM before and after the psychological interventions are reported in Appendix A, respectively. Appendix A shows the bacterial species recovered through air and surface sampling after the first five psychological interventions.

All the microorganisms were present at very low loads (CFU ≤ 1000). Potentially human-derived and truly environmental bacteria were equally represented in the pre-meeting surface sampling results (Appendix A), and account for 0.7% of the total recovered bacteria. 

At the end of the psychological sessions, potentially human-derived bacteria accounted for 66% of the findings, while truly environmental bacteria for the remaining 34% (Figure 1a,b). Among the first ones, *Staphylococcus* and *Micrococcus* were the most abundant genera (41% and 40%, respectively), followed by *Corynebacterium* (5%), *Lactobacillus* (5%), *Streptococcus* (4%), *Acinetobacter* (3%), *Rothia* (1%), and *Actinomyces* (1%) (Figure 1a). Microorganisms from *Pantoea* and *Enterobacter* genera were found, each accountable for less than 0.3% of the isolates (Appendix A). It is noteworthy that in our study we never detected *P. aeruginosa*. 

Among the environmental bacteria, Bacilli were by far the largest class (64%), followed by Actinobacteria (20%), Gammaproteobacteria (8%), Alphaprotobacteria (3%), Clostridia (3%), Betaprotobacteria (2%), and Flavobacteria (1%) (Figure 1b). Bacteria from Bacteroidia, Sphingobacteria, and Tissierellia classes were also found (each lower than 0.5% of isolates) (Appendix A).

## 4. Discussion

The aim of this study was to evaluate if subjects with CF, attending group-based psychotherapy/psychoanalytic sessions in a limited space, could contaminate the encounter-hosting environment with bacteria harbored within their respiratory tract, thus representing a source of pathogen transmission. 

Overall, our findings seem to suggest that the participation of CF patients in these meetings could be taken into consideration.

First of all, although all enrolled patients were chronically colonized with *P. aeruginosa*, we did not detect this pathogen by MEM in any of the 32 group meetings, including those in which one patient at a time was unmasked. Consistently with these findings, no changes in patient-specific *P. aeruginosa* phenotype and antibiotic susceptibility pattern were observed at the patients’ monthly microbiological follow-ups. This is particularly relevant since patient-to-patient transmission of *P. aeruginosa,* a CF leading pathogen, is described worldwide in CF care centers and is well documented with both shared and epidemic strains [9]. Furthermore, Wood et al. recently demonstrated that droplet dispersion by coughing patients is not completely halted by a mask (both surgical and N95) or cough etiquette, even if all these precautions limit the formation of airborne droplet nuclei. On the other hand, supporting our observations, they also reported that talking, as our patients did during the group meetings, is a low viable aerosol-producing activity, and that cough etiquette, observed by the patient without a mask in our study, to some extent is effective in reducing the production of aerosols containing *P. aeruginosa* (approximately 50% less effective than masks) [12].

Overall, these observations could explain the absence of viable *P. aeruginosa*, as well as other bacteria colonizing the lung patients’ cohort, in our environmental sampling.

A relative abundance of potentially human-derived bacteria was observed after psychological interventions. Gram-positive microorganisms belonging to *Staphylococcus*, *Micrococcus*, and *Corynebacterium* genera accounted for 86% of the post-meetings bacterial genera detected. These could have easily become part of the MEM-recovered microbial fall-out through the participants’ exfoliating skin and, subsequently, have persisted in the environment due to their high resistance to chemical and physical agents [21]. We can hypothesize a potential human origin also for the *Lactobacillus*, *Streptococcus*, *Actinomyces*, *Rothia*, *Moraxella*, and *Neisseria* genera, since they include commensal bacteria constituting oral microbiota, commonly found in the mucosal membranes of the upper respiratory tract, as well as in the oro- and nasopharynx. The contamination of air and surfaces (i.e., the table) could have occurred through droplet emission by the patient not wearing a mask and/or the psychotherapist. Supporting this hypothesis, such bacteria could only be found when someone was without a mask, and have not been detected in the first five meetings (Appendix A).

This observation highlights that a simple measure, such as face masks, is an efficacious strategy for infection prevention. Unfortunately, some of our patients are resistant to the use of a mask when attending the CF Center. Therefore, the results of this study suggested the implementation of an educational plan for our patients and families in order to strengthen this recommendation.

The ubiquitous Gram-negative aerobic bacillus *Acinetobacter,* which can survive up to a month on a dry surface and is commonly found on hospital worker skin (contributing to patient infection and equipment contamination), may also have had the same origin, i.e., exfoliating skin [22]. Among the several *Acinetobacter* species causing infection in humans, *A. baumannii* is responsible for almost 80% of those infections alone [22]. In our study, we detected *A. baumannii* only after one session.

Finally, Gram-negative aerobic bacilli belonging to the Enterobacter and Pantoea genera, ubiquitous in the environment (plants, soil, water, sewage) and present also on human skin and intestinal tracts [23], can be an index of environmental contamination associated with anthropic activities. 

Among the bacteria detected after the sessions, the human-derived *S. aureus* and the truly environmental *S. maltophilia* deserve separate consideration, both being potential pathogens for CF patients and both being harbored by the airways of our patients’ cohort. The first one was isolated during those sessions in which all patients wore surgical masks or when Pt3, who did not exhibit *S. aureus* lung colonization, was without a mask. As such, we can hypothesize that *S. aureus*, being a skin colonizer, became part of the microbial fall-out through the exfoliating skin [9]. A different matter concerns *S. maltophilia,* detected after group meetings in which Pt1, not having a history of *S. maltophilia* lung colonization, was without a mask. Indeed, *S. maltophilia* is commonly regarded as an environmental bacterium, and most CF people appear to independently acquire this microorganism from non-healthcare-associated environmental sources, rather than through patient-to-patient transmission [9].

Among molds and yeasts colonizing the participants’ airways, only *C. albicans* was isolated through MEM when Pt3, intermittently colonized by *C. albicans*, was without a mask. In fact, this yeast is an oral commensal microorganism and may also reside in moist areas of the skin, where it is assumed to have come from [23]. Moreover, its clinical impact on CF lung disease is uncertain.

Regarding the dynamic of the lung ecosystem in the enrolled patients over the three-year period, we observed both the persistence of the same chronic microbial species and fluctuations in transient microorganisms, except for Pt5, who showed a consolidation of *S. maltophilia* chronic colonization. It is noteworthy that we have not observed any temporal correlation between the environmental isolates, colonization changes, and patients’ interaction.

However, our study has some limitations. First, we did not perform the molecular characterization of isolates to define the clonal relationships between environmental and patient’s strains. Second, our results may not be representative for CF subjects with a different microbiological status (i.e., epidemic strains) and/or for groups with a higher number of participants (i.e., more than five)

From a psychological perspective, over time, the psychological/psychoanalytic group-based sessions allowed us to carry out transformations of the psychic contents brought by the participants, through sharing the pain and discomfort that each one was able to express, not only regarding the disease but also regarding personal painful life events that had not had expression opportunities. Feeling better from a psychological–emotional standpoint allowed patients to grow both in the management of emotions and the development of a proper ego. This was important for higher adherence to therapies, a key element in managing CF disease. 

## 5. Conclusions

Socialization between people with CF has always posed a dilemma for the CF community, who are in a continuous struggle to balance the related risks and benefits [24]. Once the role of person-to-person bacterial transmission was ascertained, those with CF have been strongly discouraged from socializing with each other, with the inevitable consequence of a lack of psychosocial support from people with shared experiences, and also through group-based psychotherapy. Conversely, our results suggest that the participation of CF people in group-based psychological/psychoanalytic sessions should be taken into consideration, as they have proven to be extremely important for the patients’ psychological status. Indeed, the precautions adopted in our setting are suitable for minimizing the risk of environmental contamination by pathogens common to the CF airways. However, protective barriers, such as surgical masks, which minimize the risk of airborne contagion, disposable gowns, a safe distance at least of 1 m, and hand hygiene, are mandatory.

## Figures and Tables

**Figure 1 ijerph-18-01142-f001:**
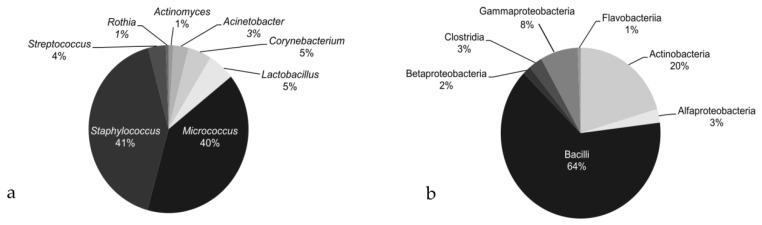
Bacteria recovered through microbiological environmental monitoring (MEM) after 32 meetings: (**a**) potentially human-derived bacterial genera; (**b**) truly environmental bacterial classes.

**Table 1 ijerph-18-01142-t001:** Microbiological status of the enrolled patients monitored over the 3-year period.

Patient	Gender	FEV1 ^1^	Colonization Year 2013		Colonization Year 2014		Colonization Year 2015	
		Range (%)	Chronic ^2^	Intermittent ^2^	Chronic	Intermittent	Chronic	Intermittent
**Pt1**	M	71–63	PA ^3^, PAM	CA, CI, KO, MSSA	PA, PAM	CA	PA, PAM	MSSA
**Pt2**	F	48–45	PA, PAM	CA, GC, KO	PA, PAM	CA, EC	Not available *	Not available *
**Pt** **3**	M	42–41	PA	CA, PAM, STM	PA, PAM	CA, MSSA	PA, PAM	AF, CA
**Pt4**	F	49–42	PA, PAM	AC, AF	PA, PAM	AF, STM	PA, PAM	none
**Pt5**	F	99–76	PA	CA, EF, PAM, MSSA	PA, PAM	AF, CA, MSSA, STM	PA, STM	none

^1^ FEV1 = forced expiratory volume in the first second. ^2^ Chronic and intermittent colonization as defined by Doring et al. [17]. ^3^ PA, *Pseudomonas aeruginosa*; PAM, mucoid phenotype *Pseudomonas aeruginosa*; AC, *Aspergillus candidus*; AF, *Aspergillus fumigatus*; CA, *Candida albicans*; CI, *Chryseobacterium indologenes*; EC, *Escherichia coli*; EF, *Enterococcus faecalis*; GC, *Geotrichum capitatum*; KO, *Klebsiella oxytoca*; MSSA, methicillin-susceptible *Staphylococcus aureus*; STM, *Stenotrophomonas maltophilia*. * The patient died at the end of 2014.

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
