# Peer review of "Environmental Microbial Contamination during Cystic Fibrosis Group-Based Psychotherapy"

_ijerph, 2021, doi:10.3390/ijerph18031142_

Round 1

Reviewer 1 Report

The manuscript describes the microbiological analysis of the environment used by 5 cystic fibrosis patients during 32 group-based psychoanalytic meetings in order to provide evidence to support these meetings when strict infection control safety precautions are taken. The mental health advantages of attending these meetings are described. The data would obviously be more powerful if additional CF patients with different microbiological status had also been sampled as mentioned in the discussion section. However, the authors do show some limited data indicating that group-based psychotherapy sessions may be considered, if infection control is maintained.

Comments:

The results of table S3 are first mentioned in the Discussion section (line 255). Data should first be presented in the Results section.

There is no explanation as to why the microbiological status of patient 2 is not available for 2015 (Table 1).

Results of the study could be further solidified if pulsed field gel electrophoresis, sequencing or spa-typing could be used to compare the various strains isolated from environmental and patient specimens to determine their relationship, especially those that are potentially pathogenic for CF patients.

Following the study, do the authors have any recommendations for improving patient-to-patient transmission prevention strategies?

The English writing of the manuscript needs to be improved. (e.g. line 22: “CF people”; line 246: “physic agents”; line 311: “socializing within each other” and “resulting into a lack of” need to be corrected)

Author Response

Dear Reviewer,

I warmly thank you for your interest in our paper.

Here below are the point-by-point responses to your valuable comments:

- I do apologize for this oversight. Table S3  is now reported in the Results section.  (lines 189-190)

- An asterix has been added to Table 1 explaining why data are not available. Additionally, an ad hoc sentence has been reported in the text. (lines 184-186)

- I agree that the results should be further solidified. We intend to do it in a similar study in the nearest future

- As for your question, I fully agree and have added a sentence accordingly. (lines 264-269)

- English writing  has  been improved.

Hoping  that  the manuscript  now meets  your requirements, I send you my best regards.

Ersilia V. Fiscarelli

Reviewer 2 Report

Authors presented a study focused on the risk of of dissemination of microorganisms in a restricted environment where CF patients attend group psychotherapy sessions. 

First I think that the study is very interesting to understand the impact of microrganism contamination during social activity of CF patients, a challenging point for the CF community. 

I have some concern about the Microbiological status of the enrolled patients monitored over the 3-year period: i suggest to add some comments about the changing that occured during the three years. 

There is a relation between the strains identified in the environment and the changing of the bacterial contamination of the patients?

Additional comments:

Table 1. Could the author comment this table? there is a correlation beetween the colonization changing and the interaction between the patients? why the informaion about patient 2 are not available in the 2015?

line-163  Which is the distance between the patient without the mask and the others?

Conclusions: I would like to know if the authors have some suggestion regarding the behaviour that must be taken during the socialization between people with CF.

Author Response

Dear Reviewer,

I sincerely thank you for your kind revision of our manuscript.

Here below are my answers to your comments:

- I fully agree with your valuable suggestions. Therefore, I have added my comments in the text (lines 304-310)

-As for your first question, I confirm there is no relation between the strains identified in the environment, the changing of the bacterial contamination of the patients, and the interaction between the patients (see lines 304-310). As for the second your question, I clarify that the distance between the patient without mask  and the others was 1 meter  at least.

- An asterix has been added to Table 1 explaining why data are not available. Additionally, an ad hoc sentence has been reported in the text (lines 184-186)

- It is authors' opinion that the study provides evidence that a meticolous and consistent adherence to simple misures as masks, gowns and gloves (and social distancing, also) is the key strategy for infection prevention. In our CF Center, an educational program on mask use for patients and families was needed to reiforce the recommendations (lines 264-269)

- English spelling  has  been revised.

Hoping to have met your requirements, I send you my best regards.

Ersilia V. Fiscarelli